# Pain Trajectory after Short-Stay Anorectal Surgery: A Prospective Observational Study

**DOI:** 10.3390/jpm13030528

**Published:** 2023-03-15

**Authors:** Yujiao Zhang, Yangyang Xia, Yue Yong, Yalan Zhou, Zhiyu Yin, Jing Wang, Ling Mei, Wei Song, Jian Wang, Jiangang Song

**Affiliations:** 1Department of Anesthesiology, Shuguang Hospital Affiliated with Shanghai University of Traditional Chinese Medicine, Shanghai 201203, Chinawj_3096@shutcm.edu.cn (J.W.); 2Research Institute of Acupuncture Anesthesia, Shuguang Hospital Affiliated with Shanghai University of Traditional Chinese Medicine, Shanghai 201203, China

**Keywords:** pain, group-based trajectory model, anorectal surgery, postoperative pain, analgesia

## Abstract

The evolution of pain after anorectal surgery has not been well characterized. The main objective of this study is to evaluate patterns in acute postoperative pain in patients undergoing short-stay anorectal surgery. A total of 217 patients were included in the study, which used group-based trajectory modeling to estimate postoperative pain and then examined the relationships between sociodemographic or surgical factors and pain trajectories. Three distinct postoperative pain trajectories were determined: hemorrhoidectomy (OR, 0.15), higher anxiety (OR, 3.26), and a higher preoperative pain behavior score (OR, 3.15). In multivariate analysis, they were associated with an increased likelihood of being on the high pain trajectory. The pain trajectory group was related to postoperative analgesic use (*p* < 0.001), with the high-low group needing more nonsteroidal analgesics. The study showed that there were three obvious pain trajectories after anorectal surgery, including an unreported low-moderate-low type. More than 60% of patients maintained moderate to severe pain within 7 days after the operation. These postoperative pain trajectories were predominantly defined by surgery factors and patient factors.

## 1. Introduction

Hemorrhoids and anorectal abscess fistulas are the two most common types of benign perianal disease with a high incidence rate [1,2,3]. It has been estimated that 25% of British and 75% of American citizens will experience hemorrhoids at some point in their lives [4]. The incidence of anal fistulas is approximately two per 10,000 individuals per year [5]. Although benign perianal diseases are not fatal, delayed or inadequate treatment could lead to chronicity, impaired quality of life, and even serious impacts [1,2,6,7], such as perianal sepsis and colorectal cancer [5,8]. Surgery is the primary treatment for these two benign perianal diseases [9]. For patients with hemorrhoids, surgical resection can relieve symptoms faster (3.9 days for resection and 24 days with conservative treatment), and postoperative recurrence is low (one-year recurrence rate, 6.4%; conservative treatment, 25.4%) [10,11]. Surgery is also the first choice for patients with anal fistulas [12], with a cure rate of >90% [6,13].

Muscle groups located around the anus have a specific autonomous ability to relax and tighten. Many sensory and motor nerves are distributed and dominate these muscles [14,15], mainly the inferior hemorrhoid, anterior sphincter, anococcygeal, and perineal branches of the fourth sacral nerve [16]. Naturally, trauma and inflammation caused by surgical treatment in this area can lead to serious postoperative pain; moreover, defecation exacerbates postoperative pain. It also causes patients to fear defecation and may lead to constipation, further aggravating the pain. In addition, patients undergoing short-stay anorectal surgery are usually discharged from the hospital two to three days after the operation. This makes postoperative pain after discharge easy to ignore by doctors, resulting in insufficient analgesia and decreased patient satisfaction [17,18]. Consequently, it is necessary to provide personalized analgesia regimens to promote patient rehabilitation and improve patient satisfaction. However, the evolution of pain after anorectal surgery has not yet been well characterized.

The main goal of the study was to characterize, through a group-based trajectory analysis, unique groups of postoperative pain trajectories for postoperative days 1–10 following surgery for benign perianal disease. The second goal was to investigate how sociodemographic and clinical factors affected the pain trajectory in patients who had surgery for with benign perianal diseases.

## 2. Methods

This trial is a prospective cohort study using samples from benign perianal surgery. The purpose is to study the relationship between population-based trajectory modeling and postoperative pain trajectory. The study protocol (institutional review board approval No.: ChiECRCT20210631) was approved by the institutional review board’s China registered clinical trial ethics review committee. All patients provided written informed consent for their clinical data to be used for research purposes; they are usually recruited and registered more than 24 h before the operation to avoid unnecessary time pressure. This manuscript adheres to the STROBE (Strengthening the Reporting of Observational Studies in Epidemiology) guidelines.

The study included patients undergoing elective surgery for benign perianal diseases, including hemorrhoids, anal fistulas, perianal abscesses, anal fissures, and rectal prolapse. The inclusion criteria were as follows: surgical indications and surgical treatment; age > 18 years; estimated hospitalization time >48 h; the expected survival time was >6 months; American Society of Anesthesiologists (ASA) Classes I–II. Individuals with other important organ diseases, those who had previously undergone anorectal surgery, experienced postoperative bleeding, infection, or other complications, and were readmitted, individuals with neurological or mental disorders, and those with communication difficulties were excluded. All subjects underwent surgery at Shuguang Hospital, affiliated with the Shanghai University of Traditional Chinese Medicine, between 22 February and 18 May 2021. Patients who fulfilled these criteria were usually screened and recruited by a trained study coordinator one to two days before surgery. In addition, all eligible patients were screened by appropriate clinical staff on all weekdays and at all hours to minimize the risk of selection bias. A flow diagram illustrating the sample selection process is presented in Figure 1.

A total of 244 patients underwent hemorrhoidectomy or anal fistula resection in the Shuguang Hospital during the study period from February to May, 2021. Of these patients, 217 were enrolled.

### 2.1. Anesthesia and Analgesia

The patients were given an oxygen mask inhalation (2 L/min) and intravenous infusions before the operation to maintain stable vital signs. Blood pressure, pulse oxygen saturation, and electrocardiography findings were monitored during the operation. All patients accepted the standardized surgical program of deepening rehabilitation, and the specific process was propofol (2 mg/kg) and sufentanil (0.2 μg/kg) for intravenous anesthesia induction. After the patient fell asleep, a bilateral pudendal nerve block was performed with ropivacaine 0.2% (AstraZeneca, Stockholm, Sweden) and epinephrine (10 μg/mL). Intraoperative maintenance was performed with propofol (1.0–2.0 mg/(kg·h)). Vital signs were closely monitored during the operation, and symptomatic treatment was administered based on the response of the respiratory and circulatory systems. The main postoperative analgesics are oral paracetamol and external suppository. Compound paracetamol tablets (Bayer, Leverkusen, Germany) are taken twice per day, with one tablet (450 mg) for each dose. If pain was severe, it was used once more (not exceeding four tablets). External suppository is the Mayinglong Hemorrhoids Suppository (Mayinglong, Huanggang, China). The main ingredients are artificial musk and artificial bezoar, which have the effect of clearing heat and relieving pain. Take one capsule (330 mg) twice a day.

### 2.2. Data Collection

#### 2.2.1. Sociodemographic and Clinical Measures

Sociodemographic variables, including age, sex, height, and weight, were obtained from the electronic health records using the values listed at the time of surgery. The preoperative pain behavior score (UBA pain behavior scale) (i.e., behaviors that would indicate to others that an individual is experiencing pain, such as grimacing or sighing) [19] and preoperative anxiety scores (PAS-7) (e.g., “I worry about the effect of surgery”) [20] were recorded. In addition, the type of surgery and the surgeon were recorded (chief physician, referring to a doctor with >10 years’ experience; attending physician, a doctor who has worked for <10 years). The duration of intravenous or oral nonsteroidal analgesic use was also recorded.

#### 2.2.2. Pain Assessment

A short pain scale was used daily for 10 days after the operation. Postoperative pain was assessed using a visual analogue scale (VAS). Subjects described the average intensity of pain per day after surgery (0 represented “no pain” and 10 represented “the highest degree of pain you can imagine”) and the frequency of taking analgesics. A trained research coordinator collected postoperative transient pain scale scores in the ward. After discharge, subjects were contacted via telephone to complete the evaluation for an additional 10 days.

#### 2.2.3. Outcome Measures

The main outcome measure was the VAS score from days 1 to 10 after the operation. Secondary outcome measures included sociodemographic data (sex, age, height, weight, body mass index (BMI), preoperative anxiety, and pain behaviors), surgery-related factors (the surgeon and the type of surgery), and the impact of postoperative analgesic drug consumption on the postoperative pain trajectory.

### 2.3. Statistical Analysis

Statistical analysis was performed using SPSS version 26.0 (IBM Corporation, Armonk, NY, USA). Continuous measurements are summarized as mean ± SD and classified as proportions. The normality of the main measures of mean pain was graphically examined using histograms stratified according to postoperative day.

In the preliminary analysis, patient groups or subgroups with similar postoperative pain progressions (i.e., trajectories) were identified, and group-based trajectory modeling was implemented in the Stata software using maximum likelihood estimation. All subjects were classified as members of a trajectory group, and the posterior probability of becoming a member was the highest [21]. Group-based trajectory modeling was used to determine the number of different trajectory groups and the shape of each trajectory (i.e., the order of polynomials). Model fitting followed a two-stage iterative process. First, each track group is modeled as a high-order shape (i.e., a cube). Then, by comparing the models of different groups, the number of groups is determined by the Bayesian coefficient and a posteriori probability. After determining the number of groups, the model was used to determine the shape of each track. The Bayesian information criterion was used to identify the simplest and best-fitting model; more specifically, the model with the least number of trajectories had the best fit [21].

In the secondary outcome analysis, the relationship between several preoperative factors and trajectory groups was examined. Categorical preoperative factors were analyzed using the chi-squared test or Fisher’s exact test, and continuous preoperative factors were analyzed using one-way analysis of variance (i.e., ANOVA). The parameter test used quantile-quantile diagrams to evaluate the normal hypothesis; if it was violated, the nonparametric Kruskal-Wallis test would be used. Levene’s test was used to evaluate the equal variance hypothesis. If the hypothesis was violated, the Welch correction was used. The pain locus was used as the basis for classification, and the factors were analyzed using multiple logistic regression. Variable selection was based on univariate criteria (*p* < 0.25) [21]. Collinearity was assessed using a univariate analysis of variable expansion factors. Estimates are reported as odds ratios (OR) and the corresponding 95% confidence intervals (CI). Differences with *p* < 0.05 were considered to be statistically significant [22].

## 3. Results

A total of 244 patients were recruited, of whom three underwent other types of surgery and 10 were lost to follow-ups. A total of 231 individuals were registered, of whom 12 had incomplete data at the time of analysis and two had no available postoperative data. As such, data from 217 participants was included in the final analysis (Figure 1).

### 3.1. Patient Characteristics

A total of 217 patients were included in the present study. Patient demographic information and the types of surgery are summarized in Table 1. The mean age of the patients was approximately 41 years, with 68.7% males and 31.3% females. The mean BMI was 24.5 kg/m^2^. Among the samples, 58.1% underwent a procedure for prolapse and hemorrhoids (PPH), 40.5% received a fistulectomy, 0.9% received seton therapy, and 0.5% received a fistulotomy. The mean preoperative anxiety and pain behavior scores were 1.3 ± 0.9 and 2.2 ± 1.0, respectively.

### 3.2. Daily Pain and Group-Based Trajectory Analysis

A total of 217 patients were used for the group-based trajectory analysis. In the entire sample, the mean daily pain score was moderate on postoperative day 1 (5.1 ± 3.1) and decreased significantly by postoperative day 10 (1.8 ± 1.0) (Figure 2). Table 2 reports the model fitting for the group-based trajectory analysis. The optimal number of trajectories and trajectory shapes are determined through group-based track modeling. The average posterior probabilities (AvePP) of the optimal model were closer to 1, and the Bayesian information criterion (BIC) value was relatively small. The best-fitting model included three trajectory groups: linear, quadratic, and cubic. These track groups are graphically illustrated in Figure 3.

Fifty-seven percent of patients were in the high-low group, 23.3% were in the low group, and 19.7% were in the low to moderate-low group.

The characteristics of patients in the trajectory group are summarized in Table 3. The percentage accumulation histogram between the surgery and track groups is shown in Figure 4. The following is a summary of the main demographic and clinical characteristics of each pain track group:

Low: The initial pain in patients with this trajectory was mild until it exhibited a downward trend. The average age of the patients was 41.8 ± 11.7 years, and the male to female ratio was approximately 6:4. The mean BMI was 25.1 ± 3.7 kg/m^2^. The proportion of patients receiving PPH and fistulectomy is almost equal; one patient received seton therapy. Nearly half of the surgeons were chief physicians with rich clinical experience. The preoperative anxiety and pain behavior scores were 0.7 ± 0.5 and 1.9 ± 1.1, respectively. The total consumption of postoperative nonsteroidal analgesics was 2.2 ± 0.5 g.

Low-moderate-low: The initial pain in this pain trajectory group was mild but gradually aggravated until it began to decrease after reaching the peak six to seven days after the operation. The mean age of the patients in this group was 39.1 ± 11.1 years, and the male to female ratio was approximately 7:3. The proportion of patients who underwent hemorrhoidectomy and fistulectomy was approximately 4:5 (19:24). The mean preoperative anxiety and pain behavior scores were 1.0 ± 0.6 and 1.9 ± 1.0, respectively. The mean total consumption of postoperative nonsteroidal analgesics was 2.7 ± 0.5 g.

High-low: The initial pain of patients in this pain trajectory group was very serious but became increasingly lighter with each day. The mean BMI was 24.1 ± 3.9 kg/m^2^. The proportion of patients who underwent hemorrhoidectomy and fistulectomy was approximately 2:1 (82:39), and one patient received seton therapy and one received a fistulotomy. Nearly 60% of the patients underwent surgery performed by an attending physician. The mean preoperative anxiety and pain behavior scores were 1.7 ± 0.9 and 2.4 ± 1.0, respectively. The mean total consumption of postoperative nonsteroidal analgesics was 3.2 ± 0.8 g.

### 3.3. Multivariable Analyses

Variables included in multiple logistic regression analyses were: BMI, types of surgery, preoperative anxiety score, preoperative VAS pain score, and postoperative analgesics. The low group was used as the reference group for multiple logistic regression analyses. Table 4 summarizes the results of the analyses, including 95% CIs. Type of surgery, preoperative anxiety score, and pain behavior score are all independent predictors of pain trajectory. Specifically, PPH (OR 0.15; 95% CI 0.02 to 0.99), higher anxiety (OR 3.26; 95% CI 1.04 to 10.26), a higher preoperative pain behavior score (OR 3.15; 95% CI 1.23 to 8.07), and more postoperative analgesics (OR 12.63; 95% CI 4.00 to 39.90) were associated with an increased likelihood of a high-low pain trajectory.

## 4. Discussion

According to group-based trajectory analyses, three pain trajectories identified patients with low, high-low, and low-to-moderate pain over time in the first 10 days after surgery for perianal benign disease. Patients who underwent hemorrhoidectomy were more common in the high-low group. Moreover, this group experienced more preoperative anxiety, exhibited pain behaviors, and had a greater demand for postoperative analgesia.

Anorectal surgery could be carried out under general anesthesia, spinal anesthesia, nerve block, local anesthesia, or combined anesthesia. A meta-analysis found that the combination of local anesthesia and intravenous sedation can significantly reduce the pain score at 6 and 24 h after surgery [23]. Therefore, there will be no complications related to spinal anesthesia, such as headache, urinary retention, etc. Nerve block is a very effective way to reduce postoperative pain, which can reduce the pain caused by the surgical incision and the inflammatory reaction of wound tissue. Therefore, we chose general anesthesia and a pudendal nerve block for anesthesia.

The pain trajectories of our high-low and low groups were similar to those of other comprehensive surgery types (including elective, major orthopedic, urological, colorectal, pancreatic/biliary, thoracic, or spine surgery) reported previously [24,25,26]. Generally, the resolution of postoperative pain was either severe or moderate at first, then gradually decreased, or the degree of pain was mild from beginning to end. Notably, the pain trajectory in our low-moderate-low group was highly novel and, to our knowledge, has not been previously reported. The initial pain was mild (the lightest of the three groups), increased to moderate on postoperative day 3, peaked on postoperative day 7, and decreased to mild on postoperative day 9. We speculate that this distinctive pain curve may be related to the special anatomical structure and function of the anus. Considering the influence of defecation after the operation and the repeated contraction and relaxation of the anal sphincter, inflammation, and edema at the wound site after three days, it may be more serious than at the initial stage after the operation, thus resulting in increased pain. On day 7, with the proliferation of wound granulation tissue, the wound line knot falls off and pain peaks. By day 9, the pain was gradually relieved as postoperative inflammation and edema subsided.

Our results also demonstrated that postoperative pain in benign perianal surgery was severe and was not effectively resolved. In our included population, more than one-half (57%) were in the high-low group. The pain score on the first day after the operation was severe (>7 points); it was stable and persisted from days 1 to 9 and decreased to mild on day 10. The low-moderate-low group accounted for nearly 20% (19.7%), and the pain increased to moderate pain (4 to 6 points) on postoperative day 3 and decreased to mild on day 9. These two types of pain accounted for nearly 80% (76.7%) of the patients, while the low group (23.3%) experienced mild pain. In summary, 62.7% of patients reported stable and sustained moderate-to-high pain over the first seven days after surgery. Moreover, although the high-low group used more analgesics, this did not effectively decrease the postoperative pain score. Recently, a prospective cohort study also indicated high postoperative pain scores despite a multimodal analgesia approach, including local anesthetic infiltration and multiple oral analgesics (paracetamol, diclofenac, and tramadol), which were applied in ambulatory anorectal surgery [18]. Thus, a better analgesic scheme merits further investigation.

Our study demonstrated that pain trajectories are related to surgical and patient factors. The number of patients who underwent hemorrhoidectomy was higher in the high-low group, consistent with our clinical experience. This may be related to the location of the occurrence. Hemorrhoids generally occur in the most sensitive areas of the rectum and anal canal, whereas anal fistulas mostly occur in the anal soft tissue outside the operation area [5,27,28]. Our regression results also revealed a correlation between postoperative pain trajectory and preoperative anxiety and pain behavior scores. The main reason for this is that higher preoperative anxiety activates the sympathetic nervous system and potentiates adrenalin and noradrenalin from the adrenal medulla, which increase the sensitivity of pain receptors and decrease the pain threshold, resulting in more intense postoperative pain [29]. Pain behaviors (e.g., complaints, painful grimaces, and changes in body posture) arouse the concern and sympathy of the individuals around them. These reactions act as a reward and enhance patient pain behavior, which leads to an increase in the intensity of pain experienced [30]. Our research also showed that patients in the high-low pain trajectory had greater preoperative anxiety, higher preoperative pain behaviors, and more analgesic drugs, indicating that anorectal patients receiving higher chronic pain for a long time before operation may be in a pain-sensitive state and more likely to cause higher postoperative acute pain and prolong its lasting. Consistent with the results of Markus et al., preoperative pressure threshold (PPT) is related to postoperative pain in short-stay anorectal surgery and may help to identify which patients have the greatest risk of more severe postoperative pain [17]. In previous studies, postoperative pain was generally reported as an average value, potentially blurring individual variations in longitudinal pain experiences and identification of patient characteristics [14,18,31,32,33]. As a new method, the pain trajectory can be used to determine subgroups [34] and understand the characteristics of susceptible patients and related risk factors for pain [21,24]. It can also quantify the speed and intensity of pain resolution and identify less-than-optimal pain relief [21]. Furthermore, pain trajectories can be customized with judicious pain management strategies after the patient’s discharge, and individualized follow-up times can be formulated [34,35].

According to our data, more attention should be devoted to patients in the low-moderate-low group in the first three days following surgery, when pain is aggravated. As such, preemptive analgesia and analgesic drugs (or treatments) should be administered to reduce pain. Furthermore, patients were educated by being informed that, excluding surgical complications or infection, aggravation of pain is a normal phenomenon and should not cause significant concern.

Our study had several limitations, the first of which was its single-centered design, which may limit the generalizability of our findings since there are significant differences in operative methods and the administration of analgesics across various hospitals. In addition, the sample size limited our ability to make stronger inferences about relevant factors influencing pain trajectories, including multivariate analysis and evaluation of longitudinal changes in pain protocols. This study only included patients undergoing benign perianal surgery under general anesthesia and nerve block. For other anesthesia methods, the postoperative pain trajectories need further study.

## 5. Conclusions

Overall, our results revealed that there were three types of postoperative pain trajectories for benign perianal diseases after the administration of general anesthesia and a nerve block. Initially, some patients exhibited special pain trajectories, such as low-moderate-low, and 62.7% of patients reported stable and sustained moderate-to-high pain over the first seven days after surgery. These postoperative pain trajectories are predominantly defined by surgery and patient factors.

## Figures and Tables

**Figure 1 jpm-13-00528-f001:**
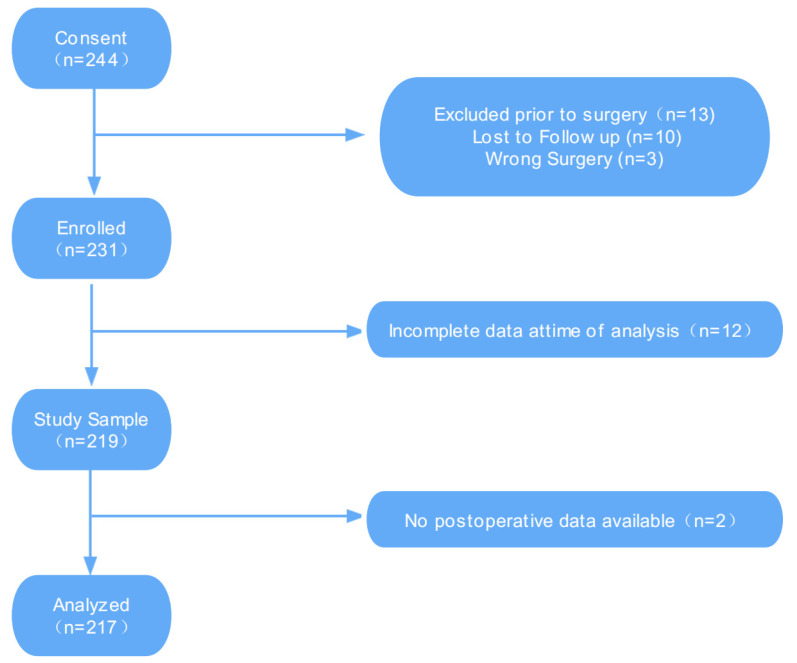
Flow diagram.

**Figure 2 jpm-13-00528-f002:**
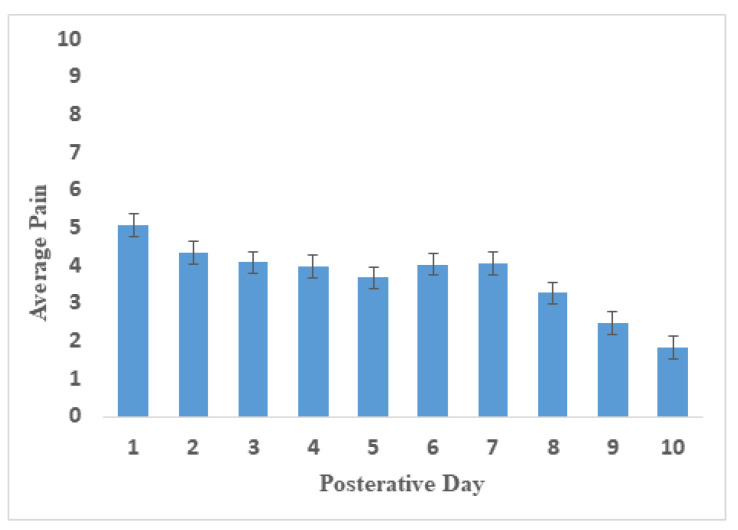
Average daily pain in the first 10 days after surgery. Abbreviations: CI, confidence interval. Average daily pain in the first 10 days after surgery. Error bars indicate 95% CIs.

**Figure 3 jpm-13-00528-f003:**
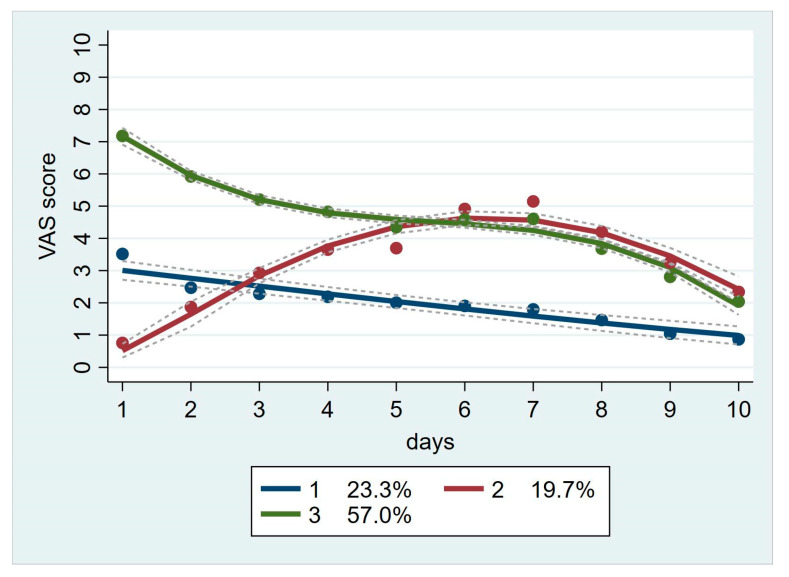
Group-based pain trajectories for the first 10 days after surgery. Pain trajectories for the first 10 days after the operation. 1 is the low group, accounting for 23.3%; 2 is the low-moderate-low group, accounting for 19.7%; and 3 is the high-low group, accounting for 57.0%. VAS indicates visual analog scale.

**Figure 4 jpm-13-00528-f004:**
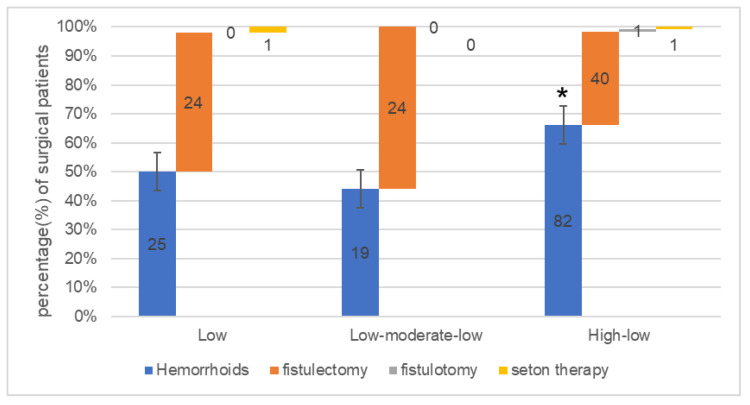
Percentage-stacked histogram of surgical and pain trajectories. The number in each column indicates the number of patients in that group. * indicates *p* < 0.05.

**Table 1 jpm-13-00528-t001:** Patient demographics.

		Frequency/Average	Percent (%)
Age, mean years ±SD		41.0 ± 12.0	
Sex	Female	68	31.3
	Male	149	68.7
Height, mean cm ± SD		168.7 ± 8.8	
Weight, mean kg ± SD		70.1 ± 14.0	
BMI, mean ± SD		24.5 ± 3.8	
Marital status	Unmarried	24	11.1
	Married	183	84.3
	Divorce	10	4.6
Type of Surgery	PPH	126	58.1
	Fistulectomy	88	40.5
	Fistulotomy	1	0.5
	Seton therapy	2	0.9
Preoperative anxiety score, mean ± SD		1.3 ± 0.9	
Preoperative pain behavior score, mean ± SD		2.2 ± 1.0	
Hospital length of stay, mean d ± SD		4.6 ± 1.8	

Abbreviations: SD, standard deviation. Data are presented as mean ± SD, median (IQR), or n (%), unless otherwise specified. BMI = body mass index.

**Table 2 jpm-13-00528-t002:** Average posterior probabilities of group assignment and Bayesian Information Criterion (BIC) statistics of trajectory model fitting.

	Group 1	Group 2	Group 3	Group 4	Group 5	BICs (*n* = 217)
2 groups	0.98	0.98				−4055.30
3 groups	0.98	0.96	0.99			−3825.37
4 groups	0.96	0.98	0.91	0.97		−3762.18
5 groups	0.97	0.97	0.93	0.95	0.97	−3706.50

Abbreviations: BIC, Bayesian Information Criterion.

**Table 3 jpm-13-00528-t003:** Patient demographics by group.

Patient Demographics		Low (*n* = 50)	Low-Moderate-Low (*n* = 43)	High-Low (*n* = 124)	*p*-Value
Age, mean, yr ± SD		41.8 ± 11.7	39.1 ± 11.1	41.4 ± 12.4	0.402
Sex	Female	18 (36%)	13 (30.2%)	37 (29.8%)	0.719
	Male	32 (64%)	30 (69.8%)	87 (70.2%)	
BMI, mean ± SD		25.1 ± 3.7	24.8 ± 3.5	24.1 ± 3.9	**0.077**
Type of Surgery	PPH	25 (50%)	19 (44.2%)	82 (66.1%)	**0.018**
	Fistulectomy	24 (48%)	24 (55.8%)	40 (32.3%)	
	Fistulotomy	0	0	1 (0.8%)	
	Seton therapy	1 (2%)	0	1 (0.8%)	
Surgeon *	Chief physician	24 (48%)	16 (42%)	48 (41%)	0.678
	Attending physicians	26 (52%)	22 (58%)	70 (59%)	
Preoperative anxiety score, mean ± SD		0.7 ± 0.5	1.0 ± 0.6	1.7 ± 0.9	**0.001**
Preoperative pain behavior score, mean ± SD		1.9 ± 1.1	1.9 ± 1.0	2.4 ± 1.0	**0.005**
Postoperative analgesics, mean, g ± SD		2.2 ± 0.5	2.7 ± 0.5	3.2 ± 0.8	**0.001**

Abbreviations: SD, standard deviation. * Indicates that some data is missing; bold indicates *p* < 0.25; and red indicates *p* < 0.05.

**Table 4 jpm-13-00528-t004:** Results from multinomial logistic regression. The low-pain group was used as the reference group for trajectory results. The odds ratio is used to represent the odds ratio of the similarity of group members per unit increase. The bold parameter represents the odds ratio of 95% CI excluding 1.

	Odds Ratio (95%CI)	
Patient Demographics	Low-Moderate-Low	High-Low
BMI, kg/cm^2^	0.99 (0.86, 1.14)	0.95 (0.83, 1.09)
Types of Surgery	**6.85 (1.00, 46.93)**	**0.15 (0.02, 0.99)**
Preoperative anxiety score	1.52 (0.49, 4.73)	**3.26 (1.04, 10.26)**
Preoperative pain behavior score	0.47 (0.18, 1.25)	**3.15 (1.23, 8.07)**
Postoperative analgesics, g	**10.61 (3.37, 33.40)**	**12.63 (4.00, 39.90)**

## Data Availability

The data presented in this study are available upon request from the corresponding authors.

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
