# Peer review of "Pain Trajectory after Short-Stay Anorectal Surgery: A Prospective Observational Study"

_jpm, 2023, doi:10.3390/jpm13030528_

Round 1

Reviewer 1 Report

Major comments: 

This is a nice manuscript about post-operative pain on anorectal surgery. Unfortunately it is a retrospettive analysis of a surgery population treated intraoperatively in a standard way…not well explained and seeming to be not enough for this surgery. This is the main limit of this study. The possibility of different anaesthesiology and analgesic pathways should be discussed. Moreover, the component of chronic pain experienced by the patients with haemorrhoids might influence a lot the acute postoperaotive pain and prolong its lasting. Also this should be discussed.

From a general point of view, the results section is a little bit redundant because all data presented in tables is also written in the text.  

Material and methods section, section 2.1: This section is important because intraoperative strategy is the base of reducing or controlling post-operative pain. It is not clear…no spinal anaesthesia? Moreover, general anaesthesia maintained with propofol continuous infusion? The dosage are not correct. Finally no analgesia started during surgery, only the nerve block and then directly tablets of paracetamol? Moreover paracetamol at the dosage of 450 mg is very low dosage for pain control. An again, this is an international journal, what is: “Supplemented with SheXiangZhiChuangShuan (MaY- 100 ingLong, China), one capsule (330 mg) was administered twice daily.” ??

Results, line 189-193 and figure 3: it is not clear from methods section how you determine the three groups. 

Minor comments: 

Line 89 mathods section: “ All patients were administered oxygen (2 L/min) via mask inhalation.” has to be re-write, not correct English. 

Line 91: “ The peripheral vein channel of the left upper limb was then opened.” what does it mean? Not important in the text. 

Line 95: “ Propofol (2 mg/kg) and sufentanil (0.2 μg/kg) were administered intraoperatively.” you used TCI anaesthesia? This is not clear, the dosage is induction dose…not propofol continuous infusion. 

Results, line 189-193: “ Pain trajectories for the first 10 days after the operation. 1 is the low group, accounting for 23.3%; 2 is the low-moderate-low group, accounting for 19.7%; 3 is the high-low group, accounting for 57.0%.VAS indicates Visual Analog Scale. Fifty-seven percent of patients comprised the high-low group, 23.3% were in the low group, and 19.7% were in the low- to moderate-low group.” re-write please, you write twice the same results.

Figure 4 legend: again you repeated the same phrase twice. Please edit. Moreover I do not see in the figure any *

Discussion:

line 224: typing error. Please edit

Reviewer 2 Report

the manuscript titled Pain trajectory after short-stay anorectal surgery: a prospective observational study is a work that demonstrated three trajectories after anorectal surgery. Interestingly they described a particular pain trajectory, the low-moderate-low group in the first three days following surgery when pain is aggravated. They indicated in the discussion that according to their data, more attention should be devoted to patients in the low-moderate-low group in the first three days following surgery when pain is aggravated. As such, preemptive analgesia should be administered, and analgesic drugs (or treatments) should be administered to reduce pain.  (but what kind of analgesic they recommended administrated to the patient, in their experience?)

The manuscript is so interesting and well-written. However, I recommend improving the quality and homogenising the same format in the figures. 

Round 2

Reviewer 1 Report

The authors have edited the manuscript based on all my concerns and now I think the manuscript is suitable for publication. No further comments.